# Financial Fraud against Older People in Hong Kong: Assessing and Predicting the Fear and Perceived Risk of Victimization

**DOI:** 10.3390/ijerph19031233

**Published:** 2022-01-22

**Authors:** Jessica C. M. Li, Gabriel T. W. Wong, Matthew Manning, Dannii Y. Yeung

**Affiliations:** 1Department of Applied Social Sciences, The Hong Kong Polytechnic University, Hong Kong, China; 2ANU Center for Social Research and Methods, Australian National University, Canberra 2601, Australia; gabriel.wong@anu.edu.au (G.T.W.W.); matthew.manning@anu.edu.au (M.M.); 3Department of Social and Behavioural Sciences, City University of Hong Kong, Hong Kong, China; dannii.yeung@cityu.edu.hk

**Keywords:** crime prevention, financial fraud, fear of crime, Hong Kong, older citizens, police

## Abstract

While the majority of studies on the fear of crime focus on the impact of violent and property crimes at the population level, financial fraud against senior citizens is often under-investigated. This study uses data collected from 1061 older citizens in the community through a cross-sectional survey in Hong Kong to examine the levels of fear and perceived risk among Chinese senior citizens toward financial fraud and the factors behind them. Logistic regression analyses were conducted to assess the explanatory power of four theoretical perspectives (vulnerability, victimization, social integration, and satisfaction with police) on fear and perceived risk of fraud victimization. The results indicate significant predictive effects of victimization experience and satisfaction with police fairness and integrity on both the fear and the perceived risk of fraud among respondents. The findings not only confirm the differential impact of theoretical explanations on these constructs but can also contribute to crime prevention policy and practice in an aging society.

## 1. Introduction

### 1.1. Elder Financial Fraud

Elder financial fraud, which is also termed as “elder financial abuse” (EFA) in this study, refers to “any form of intentional perversion of truth, or trickery deliberately practice in order to gain advantage or end, unfairly and dishonestly, in order to deprive another person of his her property, legal rights, or interests, or to induce another person to relinquish something of value, and which causes that person to suffer a loss” (p. 5, [1]). The prevalence of EFA has been examined in different societies. A meta-analytic study revealed that the fraud–scam prevalence (up to a five-year period) among older adults in the U.S. was 5.6% [2]. A survey study conducted with a clinical sample in Mainland China concluded that the prevalence of EFA was 13.6% [3]. Older people tend to become victims largely because they (a) are less likely to report and acknowledge being victims of financial fraud [4], (b) like keeping cash at home [5], and (c) have a higher risk of developing dementia in later life [6]. Notably, it was evident in a survey study conducted in Spain that those respondents who had been exposed to financial fraud had greater mental health needs and poorer health-related quality of life than those who had not [7].

Johnson (2004) divided financial fraud against older people into “financial fraud committed by strangers” and “financial exploitation by relatives and caregivers” [8]. We can also categorize fraud and scams by technique, such as scams exploiting fear (i.e., taking advantage of the victim’s fear of the depreciation of property value) and scams exploiting boredom (i.e., taking advantage of the victim’s need for excitement). The United States Senate Special Committee on Aging [9] identified the top 10 most-reported scams among the 1341 complaints by the Fraud Hotline in the year of 2019. Examples include social security scams (371 complaints) and unsolicited phone calls (123 complaints). Hong Kong was ranked first in the world in terms of life expectancy in 2020 (88.1 years for females and 82.2 for males) [10]. In 2019, among the 488 total reported cases for elderly abuse, approximately 14.5% (71 cases) were related to financial abuse. This form of abuse is the second most commonly reported form of elder abuse, following physical abuse, over the past decade in Hong Kong [11].

Other than fraud victimization, fear and perceived risk of fraud were examined in previous studies. Fear and perceived risk of crime are two correlated but distinct constructs. The former refers to the emotional aspect, while the latter concerns the cognitive aspect of the threat of crime [12]. In spite of the correlation between these two constructs, they should be measured separately [13]. To measure fear, researchers usually ask respondents to talk about their worries about victimization in the past with questions such as, “*How often did you feel afraid of becoming a victim of crime on your block in the past 6 months*?” (p. 154, [14]). To understand risk, respondents are mostly invited to estimate the likelihood of their victimization in the future. For example, “*How likely is it that you will become a victim of the following crimes sometime in the next 6 months*?” (p. 155, [14]). Nevertheless, it remains uncertain whether these two constructs are distinctive based on a sample of older adults in Hong Kong.

Although a number of recent studies have investigated financial fraud or abuse against senior citizens in Chinese societies [5,15], the factors based on criminological perspectives on this particular form of abuse are less commonly examined, discussed, and compared. This study aims to fill the above knowledge gaps in examining the factors of the fear and perceived risk of financial fraud among older people in Hong Kong using four theoretical lenses. Specifically, this study intends to gain an empirical and theoretical understanding of financial fraud against older adults in Hong Kong through (a) exploring its prevalence, (b) assessing the factors and correlates of fear of fraud victimization, (c) assessing the factors and correlates of the perceived risk of fraud victimization, and (d) comparing the predictive power of different theoretical perspectives on fear and risk constructs.

### 1.2. Perspectives

Four theoretical perspectives, namely vulnerability, victimization, social integration, and satisfaction with police are adopted in this study to understand the fear and perceived risk of fraud among older people in Hong Kong. Although previous studies have investigated the impact of various explanatory variables on the fear of crime and perceived risk of crime among individuals, none have been applied to understand stranger-perpetrated fraud against older persons in Hong Kong—a specific offence, population, and site.

#### 1.2.1. Vulnerability Perspective

The vulnerability perspective attributes victimization and fear of crime to physical vulnerability and social vulnerability. Physical vulnerability refers to a circumstance in which a person is less able to resist becoming a victim of a crime because of a lack of physical strength and competence [16]. In relation to this concept, females and older persons are expected to be more fearful of crime. Boateng [17] and Franklin et al. [16] report that older persons show higher levels of fear than their younger counterparts. Conversely, Ferraro and LaGrange [18] proposed that the relationship between age and fear of some crimes may be curvilinear. Ferraro and LaGrange [19] found that fear of crime index is lowest for those aged 55–64, highest for those aged 18–24 and slightly higher for the aged 65–74 and 75+. Physical health and depression were found to be predictive of elder financial exploitation; that is, older adults with poorer physical health and higher risk of depression experience higher victimization of financial exploitation [20]. Similarly, using a focus group of 45 stakeholders in Hong Kong, Li et al. [5] found that older individuals with declining health were suitable targets for financial fraud. In a survey study of 2472 Chinese residents in urban China, women reported a higher level of fear of crime than men, and those with a high degree of self-perceived physical strength and self-defense capabilities claimed to have less fear of crime [21]. However, another study indicated a striking similarity between men and women’s fears about crime [22]. In spite of the mixed results generated from previous studies, this study assumes a positive relationship between age, gender (female), physical health, and fear and perceived risk of fraud victimization.

Social vulnerability refers to a circumstance in which a person is less able to resist crime victimization because of his/her deficit in terms of social resources or community networks [16]. *Guanxi*—close relationships in the immediate neighborhood—has been confirmed as a significant indicator of fear of crime in contemporary urban China [23]. In a study of older adults in the U.S., Liu et al. [20] found that negative interactions with close network members and lower perceived social support significantly predicted greater financial exploitation risk. However, social network size and positive interactions with close network members were not associated with financial exploitation risk. Although the impact of social interactions on financial abuse against older persons has been demonstrated, the impact of the size of social networks remains uncertain. Additional efforts are required to examine these two variables of social vulnerability. In this study, positive interactions with friends and neighbors and the size of social networks are assumed to be positively associated with fear and perceived risk of fraud.

#### 1.2.2. Victimization Perspective

The victimization perspective assumes that “experiences of previous victimization increase one’s level of fear of future victimization” (p. 566, [17]). This assumption is generally supported by previous studies. In a study based on the analysis of the data from the International Crime Victim Surveys (ICVS), a strong link between victimization experience and fear was found across 17 industrialized countries [24]. Respondents in a study conducted in urban China, who had been victims of property crime, expressed a higher level of fear of crime [21]. Another study conducted in Ghana with 1000 residents supported the positive link between fear of crime and prior experience of victimization [17]. Nevertheless, this relationship has not been supported in other studies [25,26]. Hinkle [14] reveals that previous victimization has an impact on emotional fear but not on cognitive fear (i.e., perception of risk). In this study, other than prior victimization experience, financial loss in fraud victimization was also treated as a predictor for fear and perceived risk of being victimized for fraud.

#### 1.2.3. Social Integration Perspective

Social integration has been defined “as a person’s sense of belonging to their local surroundings as well as their attachment to the community” [16]. Fear of crime is assumed to decline with better integration within the local community through social ties with neighbors, emotional attachment to the community [27], participation in formal organizations [28], and involvement in crime watch programs [29]. Such participation or engagement not only improves knowledge of individuals in crime prevention but also creates a sense of efficacy, community attachment, and neighborhood support. Some earlier studies found a negative relationship between levels of social integration and fear of crime [27,30]. Following the conclusion of previous studies, the current study assumes a negative association between emotional fear of crime and social integration. Apropos the prediction of social integration on cognitive fear (i.e., perceived risk), the result can be polarized. For example, an individual may believe being a victim is less likely if she/he can have better skills for self-protection through his/her participation in community activities. Conversely, an individual may become more aware of his/her own vulnerability to crime if he/she is notified of victimization cases through his/her participation in these activities. In this study, a positive relationship between social integration and perceived risk of fraud victimization among older people is expected.

#### 1.2.4. Community Policing Perspective

The police play a key role in the fear of crime among citizens. Dietz [31] articulated that a “reduction of fear of crime has been associated with community policing since their inception” (p. 98). Scheider, Rowell and Bezdikian [32] offered some reasons for this. First, the increased police presence following the implementation of community policing can reduce crime and fear. Second, community policing efforts involve addressing physical and social incivilities that are often linked to increasing levels of fear of crime. Third, one of the foci of community policing is building a strong community partnership, and this partnership can enhance the feelings of efficacy among citizens, thus reducing fear of crime. Some studies find a significant relationship between fear of crime and positive attitudes toward the police [17,30,33], whereas others do not [25,32]. In summary, the positive and interactive relationship between satisfaction of police and fear of crime is well supported empirically. However, it remains unclear which dimension of police satisfaction can predict emotional fear and cognitive fear of crime. Thus, an empirical assessment is required to examine whether process-based satisfaction (police fairness and integrity) or outcome-based satisfaction (police effectiveness) has more predictive power over fear and perceived risk. In this study, we assume that both dimensions of police satisfaction have a positive impact on older persons’ fear and perceived risk of fraud victimization.

It is clear from the previous literature that fear and perceived risk of fraud among the older population can be explained through vulnerability, victimization, social integration, and community policing perspectives. However, no study to date has specifically compared the predictive power of these four theoretical perspectives with regard to older persons’ fear and perceived risk of fraud victimization. In addition, earlier studies tend to analyze fear of crime without fully appreciating the difference between the emotional and cognitive aspects of fear among the elderly. To fill this void, the current study will assess, predict, and compare both types of victimization fear from the four abovementioned perspectives.

## 2. Materials and Methods

### 2.1. Sampling Techniques and Data Collection Procedure

This study interviewed 1061 citizens aged 65 or above in the community. Given the limited representation of other racial/ethnic groups in the study site, the present study focused exclusively on people of Chinese origin. A multistep sampling method was employed. It is expected that this sampling method can help minimize the distribution bias by controlling the community characteristics, service unit natures, and respondent characteristics. Step 1 involved the selection of districts. The 19 districts of Hong Kong were divided into four types of communities according to the household income of the residents and the community fraud crime rate. For each type of community, one or two districts were randomly selected. Step 2 involved the selection of service providers. Social service providers for physically active older adults and physically inactive ones in the selected districts were invited to distribute a letter of invitation to potential interviewees. A total of 23 out of 103 prospective service providers agreed to collaborate, which accounted for a response rate of 22.3%. In Step 3, individual participants were invited to take part in the study. In this step, we sent invitations to all eligible respondents through the above service providers. A total of 23,884 invitations were sent. Among these individuals, 1093 expressed their willingness to participate (response rate of 4.6%). The final sample comprised 1061 respondents after excluding 32 incomplete responses. Data were collected with a survey questionnaire that was developed on the basis of a literature review, validated by an expert panel and finally tested in a pilot survey on 50 elderly Chinese people. A team of trained interviewers conducted face-to-face interviews with consenting older persons using a questionnaire composed of approximately 90 items. Each interview was completed in 60 min and conducted in a social service center. To increase the incentive for participation, a cash coupon worth HKD 100 (approximately USD 13) was offered to each respondent who attended the interview.

### 2.2. Participants

Table 1 provides summary statistics regarding the 1061 respondents. Nearly three quarters (73.8%) of our participants were female, with ages ranging from 65 years to over 90 years. The majority were aged between 70 and 89 years (approximately 80%). Over one-third of the respondents did not receive formal education, and only 4.5% received tertiary education. The remaining respondents either received primary education (39.8%) or secondary education (19.1%). The monthly income of the respondents ranged between less than HKD 1000 (approximately USD 128) to greater than or equal to HKD 20,000 (approximately USD 2560). Regarding living arrangements, approximately half of the sample lived alone (46.9%). In terms of the level of activities of daily living (ADL), which measure fundamental skills typically needed to manage self-care activities (e.g., grooming/personal hygiene, dressing, toileting/continence, transferring/ambulating, and eating), 93% of our respondents classified themselves as ‘without any impairment’ (i.e., Level 1).

### 2.3. Measures

The current study is intended to confirm whether the two outcome variables: (a) fear of fraud victimization and (b) perceived risk of fraud victimization are predicted by five domains of variables: (i) demographics, (ii) vulnerability, (iii) victimization, (iv) social integration, and (v) satisfaction with the police. On the basis of the previous work by Johnson (2004) and observation of the crime trend in Hong Kong, 13 types of financial exploitation against older persons *by strangers* were compiled in a list, which includes: (1) prizes and sweepstakes; (2) investments; (3) charity contributions; (4) home and automobile repairs; (5) loans and mortgages; (6) health, funeral, and life insurance; (7) health remedies; (8) travel scam; (9) employment scam; (10) phone scam; (11) burglaries or scams by disguised fraudsters; (12) spiritual blessing scams; and (13) perfume scams.

*Outcome variables.* In this study, the respondents were asked to rate their fear and perceived risk of the 13 aforementioned types of fraud. With reference to the previous study by Chadee and Ditton [34], which measured the fear index of victimization of personal and property crimes, the current study invited respondents to report their *current fear* and *perceived risk*
*of future victimization* against the 13 aforementioned types of fraud. Hence, two dichotomized variables were generated. First, a measure of *any fear* was determined; that is, whether respondents were fearful of being a victim of any stranger-perpetrated fraud. Second, a measure of *any perceived risk* was determined; that is, whether respondents perceived any risk of being a victim of any stranger-perpetrated fraud in the future.

*Predictive variables*. Control and independent variables were divided into five groups.

*Demographic variables*. The survey interview gathered information about *gender* (male vs. female), *age* (over 80 years or not), *e**ducation* (no school vs. primary education and no schooling vs. secondary or above), and *number of children* (having two or more children or not) with dichotomous responses (0 = no; 1 = yes). *Seven levels of monthly income* were included, ranging from <HKD 1000 to ≥HKD 20,000.

*Vulnerability.* With respect to measuring physical vulnerability, the respondents were differentiated by whether they had ADL dependency; that is, at least one ADL could not be performed independently (0 = no impairment; 1 = some forms of impairment). For measuring relative health, respondents were asked to compare their health with their peers in terms of physical health (0 = equal or better; 1 = worse). For measuring social vulnerability, respondents were asked if they were living alone, with a response of either no (=0) or yes (=1). They were also asked if they engaged in activities, namely, “*having four or more friends/relatives [the cut-off (4 friends/relatives) represents the median number of friends” or “relatives with frequent contact with respondents in the current study]*”, *“socializing with friends more than once a month*”, and “*chatting with neighbors more than one to three times a week*”, with responses of no (=0) or yes (=1).

*Victimization experiences.* The *number of fraud victimization experiences* in the lifetime of an individual was collected with a response of no exposure, low exposure (between one to three events), or high exposure (four or more events). Respondents were differentiated by whether they had low exposure to victimization (0 = no victimization event; 1 = one to three victimization events) and high exposure to victimization (0 = no victimization event; 1 = four or more victimization events). Respondents were also invited to indicate *any financial loss* in the most recent victimization experience with a response of no (=0) or yes (=1).

*Social integration*. Respondents were invited to offer their dichotomous responses (no = 0 or yes =1) on the various levels of their community activities over the past 12 months. These activities included “*being a member of the elderly center*”, “*being a member of the neighborhood watch*”, “*watching any crime prevention program*”, and “*participating in any crime prevention seminar or activity*”.

*Satisfaction with the police*. Two dimensions of police satisfaction (i.e., police effectiveness and police fairness and integrity) were measured using a four-point Likert scale [35]. Response categories ranged from 1 = strongly disagree to 4 = strongly agree on five questions regarding satisfaction with police effectiveness (with α of 0.82), such as “*police respond quickly to calls for help and assistance*” and on three questions regarding satisfaction with fairness and integrity (with α of 0.68), such as “*police are honest*”.

### 2.4. Analytic Plan

The analyses were conducted with SPSS software (version 26) (IBM Corp, Armonk: NY, USA) for descriptive statistics and two logistic regressions. Descriptive statistics consisted of the frequencies of variables included in the sample (see Table 1). Logistic regression analyses were performed separately for each outcome variable, with five blocks of predictive variables, namely demographic variables, vulnerability, victimization experiences, social integration, and satisfaction with the police. The Hosmer–Lemeshow logistic regression test was used to check the goodness of fit. The results of the regression analyses are presented as odds ratios (OR) with 95% confidence intervals (CIs).

## 3. Results

### 3.1. Victimization of Fraud

Table 1 shows that approximately 56% of the interviewed older individuals had been the victim of fraud at some point in their life. Among those who had been victimized, 381 (or 35.9%) encountered no more than three events of fraud (i.e., low exposure), and 204 (or 19.2%) encountered more than three events (i.e., high exposure). In this study, 26.8% of respondents claimed to have experienced fraud victimization initiated by strangers within 12 months before the interview. Among the fraud victims, 133 people (22.4%) had experienced monetary loss in their most recent fraud event. Table 2 indicates that the most common type of fraud experienced by the respondents were phone scams (27.6%).

### 3.2. Predicting Fear of Fraud Victimization

Table 3 presents the results of the logistic regression on fear of fraud. Among different demographic factors, only gender was significantly predictive of fear of fraud. Female respondents (OR = 1.99) compared with male respondents were more likely to be afraid of any fraud victimization. However, no significant association was found between fear of fraud and physical or social vulnerability. Recent victimization experiences, irrespective of level (low OR = 1.42 or high level OR = 2.15), would increase respondents’ likelihood of reporting fear of fraud. Those who had financial loss in the recent fraud event (OR = 2.12) were more likely to be fearful about this particular type of crime. With regard to social integration through community participation, individuals who participated in any crime prevention seminar or activity in the past 12 months (OR = 1.39) were more likely to report being fearful of fraud victimization. Individuals who were more satisfied with police fairness and integrity (OR = 0.82) demonstrated a decreased likelihood of being fearful of fraud. The overall model of fear explains 11.7 percent of the variance. The victimization perspective demonstrates stronger predictive power of older adults’ fear of financial exploitation by strangers, accounting for 4.7 percent of the variance in this model (see Table 3).

### 3.3. Predicting Perceived Risk of Fraud Victimization

Table 3 presents the OR of the relationship between perceived risk of fraud and theoretically inspired factors. In contrast to the model of fear, none of the demographics were found to be associated with the respondents’ perceived risk of fraud. Similar to the model of fear, vulnerability factors did not demonstrate a significant impact on predicting respondents’ perceived risk of fraud. The presence of victimization experience at low level (OR = 1.61) and high level (OR = 2.27) increased the likelihood of reporting being at risk of fraud. However, financial losses experienced in a recent victimization event revealed no impact with regard to the outcome. With regard to social integration through community participation, individuals who were a member of an elderly center (OR = 0.42) reported a lower perceived risk of victimization when compared to those who did not have such a membership. Data indicate that respondents who were satisfied with police fairness and integrity (OR = 0.79) were less likely to perceive a risk of fraud victimization by strangers. Conversely, those who were satisfied with police effectiveness (OR = 1.13) were more likely to perceive a risk of fraud victimization. Overall, all theoretical factors in this study explain 9.3 percent of the variance for the risk perception of fraud. Policing perspective compared with other theoretical perspectives demonstrates stronger predictive power for older adults’ perception of risk of financial fraud by strangers, accounting for 2.8 percent of the variance explained in this model (see Table 3).

## 4. Discussion and Implications

The objective of this study was to confirm and compare the predictors of fear and perceived risk of financial fraud towards the elderly on the basis of the four selected theoretical lenses. First, consistent with previous studies [21,36], female respondents had a higher level of fear toward being victimized than their male counterparts. This echoes the concept of “differential sensitivity to risk” by Warr [37], in which women are more likely to judge potential victimization as serious because they make subjective linkages among offenses. Meanwhile, it is speculated that male respondents under the influence of masculinity are reluctant to disclose their fear of crime to others (i.e., the interviewer). In this study, the predictive power of age (80+) on fear of fraud was marginally significant. Data indicate the respondents aged 80 or above are less likely to worry about fraud victimization compared with their younger counterparts (aged 65 to 79). This result is different from that of a previous study by Ferraro and LaGrange (1992) that found that old-old groups (75+) had slightly higher levels of fear of crime than the young-old group (65–74). Therefore, the fear of fraud victimization among the group aged between 65 and 79 in Hong Kong warrants our attention.

Second, none of the variables of the vulnerability perspective demonstrated a significant impact on an individual’s fear or perceived risk of fraud. There are four possible explanations for this outcome. First, over 93% of respondents rated their impairment of activities of daily living at the first level, and over 80% did not perceive worse health than their peers at a similar age. The respondents were predominantly healthy. Physical vulnerability may not have made sense to most respondents in this study who were active participants in the social service units. Second, regarding neighborhood support, Hong Kong (our research site) is one of the most densely populated places in the world (with 6690 persons per square kilometer). Thus, frequent interactions with neighbors are inevitable. Social vulnerability may not have mattered to respondents in this study, who were recruited through the elderly social service units. Third, the study may have omitted the cognitive and psychological aspects of vulnerability. A recent study concluded that older adults with mild cognitive decline had higher scam vulnerability whereas those with moderate to severe cognitive decline had lower fraud vulnerability [38]. Another study demonstrated that older victims of fraud compared with non-victims were more likely to exhibit severe depressive symptoms [39]. Therefore, the examination of the linear and non-linear correlations between fraud victimization, vulnerability, fear, and cognitive functioning and psychological characteristics warrants future research. Last, we share the view from the literature [38,39] that the victimization of older adults with significant cognitive impairment is typically under-reported due to their deficit in recognizing and understanding what a scam attempt is.

Third, our results support the notion that recent victimization experience increased the likelihood of older people’s fear and perceived risk of fraud. Victims are more fearful and cautious than non-victims. Victims who have experienced monetary loss are more worried than those without monetary loss. Thus, victims may require some help, for example from social workers and counsellors, to alleviate their concerns. The data also indicate that victimization experience can increase older adults’ perception of risk. This group of the population is expected to be more motivated to receive crime prevention education. Thus, timely education to victims should be promoted throughout the community. In this regard, Internet-based educational services to senior citizens are regarded as a viable option [40].

Fourth, our findings indicate that respondents who had participated in a crime prevention seminar and activities in the previous 12 months were likely to have a higher level of fear of fraud. In most crime prevention seminars in Hong Kong, the victim’s story is presented to the audience for educational purposes. This finding demonstrates the impact of “vicarious victimization” [14,41]—hearing victimization stories of older people from others in this study. However, it remains a point of debate as to whether being conscious of potential victimization is beneficial or harmful to older people’s quality of life and personal safety. On the one hand, older people may be motivated to put more effort into protecting themselves from possible victimization of fraud when they are more fearful and mindful of its risks and harm. Conversely, such knowledge may have an adverse effect. In this regard, older people may need guidance from helping professionals (e.g., social workers and community nurses) on how to handle the vicarious experience of crime.

Lastly, the results of the study suggest that older adults’ satisfaction with police fairness and integrity can help to reduce their emotional fear and perceived risk of fraud victimization. Respondents’ satisfaction with police effectiveness can mitigate their perceived risk of fraud victimization. It is suggested that process-based policing (i.e., focusing on fairness and integrity) is more important than outcome-based policing (i.e., effectiveness and efficiency) in dealing with older people’s emotional fear of fraud victimization. Police procedural justice and empathizing processes with the public are critical to day-to-day police–public encounters in Chinese societies [42], and this provides implications for police training in Hong Kong.

## 5. Conclusions

Several limitations associated with this study should be noted when interpreting the findings reported above. First, the concept of vulnerability examined in this study is limited to physical and social aspects, and other aspects of vulnerability remained unexplored. Recent literature has highlighted the important link between cognitive ability and financial exploitation against older people [43]. Second, the cross-sectional data generated from this study may make a causal inference difficult. For instance, whether the respondents had a lower level of fear due to their social integration through community participation or whether they became active in community activities due to fear of crime is hard to determine. Third, the study was conducted based on a community sample and the respondents were referred by social workers. Thus, it did not include the perspective of older adults in the residential setting and non-social service recipients. Notably, this group of elderly people may be confronted by another set of risk factors of fraud, for example, loneliness and social isolation. Fourth, the self-reported responses may contain potential biases of possible distortion in memory, mental state, and intention to offer socially desirable responses. For example, male respondents in this study may claim to have a lower level of fear than they actually experience in order to present a strong and tough image in front of others. This speculation is supported by a previous study in which men were found to be more vulnerable to fraud but less likely to report actual victimization [38]. Lastly, no specific attention being given to digital fraud seems to be a limitation in this study. With the fast-growing number of internet and mobile phone users, the possible encounters between vulnerable targets (including elderly people) and scammers are also increasing. Internet users and consumer fraud has drawn the attention of researchers [44]. In future research, a more comprehensive framework with the incorporation of digital and non-digital fraud and cognitive factors such as age-related dementia, memory, and executive functioning [45] is worth considering. Furthermore, longitudinal designs can help test causal relationships between fear of fraud and its determinants, including social integration and different types of vulnerability, by testing their temporal precedence. The perspective of older adults in different circumstances, such as those living in the community, patients staying in a residential setting and those living with cognitive decline could be incorporated to generate a more comprehensive picture of the topic.

Despite these limitations, these findings expand our knowledge regarding fear and the perceived risk of crime among the older population of Hong Kong. To our knowledge, this is first empirical study to examine and compare the predictive power of the four above-described theoretically driven variables for older people’s fear and perceived risk of financial fraud in Hong Kong. Comparatively, the victimization and policing perspectives demonstrate a greater contribution to our understanding of this specific topic in the target population. Conceptually, the current study demonstrates that fear and perceived risk of fraud are two distinctive constructs. For instance, some respondents in this study may have a high level of fear but low perceived risk, while some may not exhibit great fear but believe that they are at great risk.

## Figures and Tables

**Table 1 ijerph-19-01233-t001:** Descriptive statistics (*N* = 1061).

Variables	*N*	%
Gender		
Male	278	26.2
Female	783	73.8
Age		
65–69	137	13.0
70–74	180	17.0
75–79	222	20.9
80–84	294	27.7
85–89	155	14.6
90 or above	73	6.9
Education level		
No schooling/pre-schooling	388	36.6
Primary	422	39.8
Secondary	203	19.1
Tertiary	48	4.5
Monthly income (in Hong Kong Dollars)		
<1000	30	2.8
1000–1999	251	23.7
2000–2999	247	23.3
3000–4999	358	33.7
5000–9999	133	12.5
10,000–19,999	30	2.8
≥20,000	8	0.8
Living arrangements of the elderly		
Living alone	493	46.9
Living with spouse	238	22.6
Living with son or daughter	182	17.3
Living with spouse and children	91	8.7
Living with other relative or domestic helper	47	4.5
Level of activities of daily living impairment		
Level 1	987	93.0
Level 2	61	5.7
Level 3	5	0.5
Level 4	8	0.8
Relative health when compared with peers		
Much worse	21	2.0
Worse	172	16.2
Similar	395	37.2
Better	361	34.0
Much better	111	10.5
Fraud victimization experiences		
Recent victimization		
No	777	73.2
Yes	284	26.8
Any monetary loss in the most recent event		
No	462	77.6
Yes	133	22.4
Lifetime victimization		
No	466	43.9
Yes	595	56.1

**Table 2 ijerph-19-01233-t002:** Number and percentage of victims by type of fraud (*N* = 1061).

Victimization by Type of Fraud.	*N*	%
Prizes and Sweepstakes	137	12.9
Investments	59	5.6
Charity contributions	62	5.8
Home and automobile repairs	69	6.5
Loans and mortgages	117	11.0
Health, funeral, and life insurance	51	4.8
Health remedies	131	12.3
Travel scam	29	2.7
Employment scam	73	6.9
Phone scam	293	27.6
Home visiting scam by disguised fraudster	66	6.2
Spiritual blessing scams	40	3.8
Drug-laced perfume scam	24	2.3
Others	50	4.7
Any type of fraud	595	56.1

Each respondent could report more than one type of fraud.

**Table 3 ijerph-19-01233-t003:** Logistic regression predicting fear and perceived risk of fraud by strangers (*N* = 1061).

	Fear				Perceived Risk			
Variable	OR	(95% CI)	Adjusted R^2^	∆R^2^	OR	(95% CI)	Adjusted R^2^	∆R^2^
Demographic variables			0.045	0.045			0.01	0.01
Female	1.99 *	(1.44–2.75)			1.06	(0.77–1.45)		
Age	0.78 ^	(0.58–1.04)			0.89	(0.67–1.18)		
Primary Education	1.11	(0.81–1.52)			1.23	(0.91–1.68)		
Secondary Education	0.81	(0.55–1.20)			1.09	(0.74–1.59)		
Monthly Income	0.96	(0.86–1.08)			1.07	(0.95–1.20)		
Two or more children	1.05	(0.77–1.43)			1.22	(0.90–1.66)		
Vulnerability			0.048	0.003			0.015	0.005
ADL impairment	1.43	(0.84–2.46)			0.91	(0.54–1.54)		
Health compared with peers	1.10	(0.78–1.57)			0.92	(0.66–1.30)		
Living alone	0.75 ^	(0.57–1.00)			1.08	(0.82–1.42)		
Having friends/relatives	1.09	(0.83–1.43)			0.87	(0.66–1.13)		
Friends	0.96	(0.73–1.26)			0.83	(0.63–1.08)		
Neighbors	0.95	(0.72–1.25)			0.98	(0.75–1.29)		
Victimization			0.095	0.047			0.042	0.027
Victimization at a low level	1.42 *	(1.05–1.93)			1.62 *	(1.20–2.19)		
Victimization at a high level	2.15 *	(1.46–3.16)			2.28 *	(1.58–3.28)		
Financial loss	2.12 *	(1.33–3.39)			1.03	(0.69–1.54)		
Social integration			0.099	0.004			0.065	0.023
Being a member of elderly center	0.86	(0.53–1.41)			0.43 *	(0.26–0.69)		
Being a member of neighborhood watch	1.06	(0.75–1.49)			0.87	(0.62–1.21)		
Watching crime prevention programs	1.23	(0.59–2.55)			2.08 ^	(0.97–4.48)		
Participating in crime prevention activities	1.39 *	(1.01–1.91)			0.92	(0.67–1.27)		
Satisfaction with the police			0.117	0.018			0.093	0.028
Police fairness and integrity	0.82 *	(0.74–0.91)			0.80 *	(0.72–0.88)		
Police effectiveness	1.08 ^	(1.00–1.17)			1.13 *	(1.05–1.23)		

OR = odd ratio; CI = confidence interval; * *p* < 0.05; ^ *p* < 0.1., Age = <80 vs. ≥80; Primary education = No vs. Primary; Secondary education = No vs. ≥Secondary; Monthly Income = 7 levels ranging from <HKD 1000 to ≥HKD 20,000; ADL = Activities of Daily Living impairment = independent vs. some form of ADL impairment; Health compared with peers = equal, better or worse; Having friends/relatives = having 4 or more friends; Friends = socializing with friends more than once a month; Neighbor = chatting with neighbors more than one to three times a week; Victimization at low level = No vs. Low; Victimization at high level = No vs. High; Police fairness and integrity = measured using 3 items with a 4-point scale; Police effectiveness = measured using 5 items with a 4-point scale.

## Data Availability

Materials and anonymous data are available from the authors upon request.

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
