# Peer review of "Financial Fraud against Older People in Hong Kong: Assessing and Predicting the Fear and Perceived Risk of Victimization"

_ijerph, 2022, doi:10.3390/ijerph19031233_

Round 1
Reviewer 1 Report
I want to congratulate the authors for what I consider to be a great piece of applied research.
The issue is highly relevant in a world with an increasingly aging population and where financial crimes has more and more importance.
The work is very clear, well-structured and written. The empirical analysis is well done, and the results clearly conveyed.
Therefore, I consider that the paper is of great interest to IJERPH. There are only a couple of questions that I wanted the authors to consider. First, a minor formal question. In lines 32-4 there are a couple of errors in the numbering of “Older people tend to become victims 32 largely because they (a) are less likely to report and acknowledge being victims of finan-33 cial fraud [4], (f ) like keeping cash at home [5], and (g) suffer from dementia [6]. "
On the other hand, I miss in the study an approach to the phenomenon of digital fraud. I understand that the study does not address it specifically, but it is rare that there is not, at least some mention, or that it is recognized as a limitation. This makes the study less relevant and interesting. More having into consideration the fast growth in the use of mobile phones and internet among seniors. I would deal with this question in the exposition of the study and recognize it as a limitation of the current study and a line of further analysis.
Finally, the authors address the limitation posed by causality. I would ask them to explain what would be the best way to address its treatment in future works based on this type of survey.
Author Response
Responses to Reviewer: 1
I want to congratulate the authors for what I consider to be a great piece of applied research.
The issue is highly relevant in a world with an increasingly aging population and where financial crimes has more and more importance.
The work is very clear, well-structured and written. The empirical analysis is well done, and the results clearly conveyed.
Thank you.
Therefore, I consider that the paper is of great interest to IJERPH. There are only a couple of questions that I wanted the authors to consider. First, a minor formal question. In lines 32-4 there are a couple of errors in the numbering of “Older people tend to become victims 32 largely because they (a) are less likely to report and acknowledge being victims of finan-33 cial fraud [4], (f ) like keeping cash at home [5], and (g) suffer from dementia [6]. "
Well noted. Correction was made accordingly. We put:
“Older people tend to become victims largely because they (a) are less likely to report and acknowledge being victims of financial fraud [4], (b) like keeping cash at home [5] , and (c) often suffer from dementia [6].” (see page 1).
On the other hand, I miss in the study an approach to the phenomenon of digital fraud. I understand that the study does not address it specifically, but it is rare that there is not, at least some mention, or that it is recognized as a limitation. This makes the study less relevant and interesting. More having into consideration the fast growth in the use of mobile phones and internet among seniors. I would deal with this question in the exposition of the study and recognize it as a limitation of the current study and a line of further analysis.
Thanks for the reviewer’s suggestion. In the revised manuscript, we recognized this limitation. We added:
“Lastly, no specific attention being given to digital fraud seems to be a limitation in this study. With the fast-growing number of internet and mobile phone users, the possible encounters between vulnerable targets (including elderly people) and scammers are also increasing. Internet users and consumer fraud has drawn the attention of researchers [43]. In future research, a more comprehensive framework with the incorporation of digital and no-digital fraud and cognitive factors such as age-related dementia, memory, and executive functioning [44] is worth considering..” (see page 11).
Finally, the authors address the limitation posed by causality. I would ask them to explain what would be the best way to address its treatment in future works based on this type of survey.
“Furthermore, longitudinal designs can help test causal relationships between fear of fraud and its determinants, including social integration and different types of vulnerability, by testing their temporal precedence. The perspective of older adults in different circumstances, such as those living in the community, patients staying in a residential setting and those living with cognitive decline could be incorporated to generate a more comprehensive picture of the topic.” (see page 11).
Reviewer 2 Report
- The article is intended to be solidly constructed (strengths and weaknesses, limitations are also addressed)
- Literature Review is poor (it is normal because there are just few studies focused strictly on such a subject), but the Methodology (Materials and Methods) section is well done
- As the authors stated in the article, the paper can be a preamble for a more in-depth study, which to offer effective measures / solutions for the subject approached
Author Response
Responses to Reviewer 2
Comments and Suggestions for Authors
The article is intended to be solidly constructed (strengths and weaknesses, limitations are also addressed).
Thank you for the reviewer’s comments.
- Literature Review is poor (it is normal because there are just few studies focused strictly on such a subject), but the Methodology (Materials and Methods) section is well done.
We agree with the point that only a few studies focused strictly on this topic. We did another round of literature review and added some updated and relevant citations, including
Lichtenberg, P.A.,Sugarman, M.A., Paulson, D., Ficker, L.J., & Rahman-Filipiak, A. (2016). Psychological and functional vulnerability predicts fraud cases in older adults: results of a longitudinal study. Clinical Gerontologist, 39(1), 48-63.
https://www.tandfonline.com/doi/abs/10.1080/07317115.2015.1101632?journalCode=wcli20
Mears, D.P., Reisig, M.D., Scaggs, S., & Holtfreter, K. (2016). Efforts to reduce consumer fraud victimization among the elderly: the effect of information access on program awareness and contact. Crime & Delinquency, 62(9), 1235-1259. https://journals.sagepub.com/doi/abs/10.1177/0011128714555759
Ueno, D., Daiku, Y., Eguchi, Y., Iwata, M., Amano, S., Ayani, N., Nakamura, K., Kato, Y., Matsuoka, T., & Narumoto, J. (2021). Mild cognitive decline is a risk factor for scam vulnerability in older adults. Frontiers in Psychiatry, 12, 685451.
https://www.frontiersin.org/articles/10.3389/fpsyt.2021.685451/full
Segal, M., Doron, I., & Mor, S. (2021). Consumer fraud: older people’s perceptions and experiences. Journal of Aging & Social Policy, 33:1-21. https://www.tandfonline.com/doi/abs/10.1080/08959420.2019.1589896?journalCode=wasp20
- As the authors stated in the article, the paper can be a preamble for a more in-depth study, which to offer effective measures / solutions for the subject approached
This point is well taken. We elaborate the possible future in-depth study below:
“Third, the study may have omitted the cognitive and psychological aspects of vulnerability. A recent study concluded that older adults with mild cognitive decline had higher scam vulnerability whereas those with moderate to severe cognitive decline had lower fraud vulnerability [37]. Another study demonstrated that older victims of fraud compared with non-victims were more likely to exhibit severe depressive symptoms [38]. Therefore, ….” (see page 10)
We also enriched the parts of practice implications as follows:
“The data also indicate that victimization experience can increase older adults’ perception of risk. This group of the population is expected to be more motivated to receive crime prevention education. Thus, timely education to victims should be promoted throughout the community. In this regard, Internet-based educational services to senior citizens are regarded as a viable option [39].” (see page 10)
“Police procedural justice and empathizing processes with the public are critical to day-to-day police–public encounters in Chinese societies [41], and this provides implications for police training in Hong Kong.” (see page 11)
Reviewer 3 Report
This study examined the factors associated with fear and perceived risk of financial fraud against older people. The financial fraud against older people is one of the most important problems because the number of older people is increasing especially in Asia. Therefore, this study is valuable. However, I have some comments.
The introduction is relatively long. Authors had better describe briefly in introduction. On the other hand, the discussion is relatively short. There are few citations in the discussion and implications.
I think that fear of financial fraud prevents the fraud victimization because people with high fear of financial fraud behave more carefully. In this study, female was more likely to be afraid of any fraud victimization than male. On the other hand, a recent study showed that male was more vulnerable to financial fraud (Ueno D, et al., Mild Cognitive Decline Is a Risk Factor for Scam Vulnerability in Older Adults. Front. Psychiatry, 20 December 2021. https://doi.org/10.3389/fpsyt.2021.685451.2021). Authors had better discuss the relationship between fear of financial fraud and vulnerability to financial fraud.
The response rate of this study is low. Therefore, the participants in this study might not represent older people in Hong Kong. Please discuss this point.
I think that mental state, especially anxiety and depressive symptoms, affects the fear of fraud victimization and perceived risk of fraud victimization. Please discuss this point.
In introduction, authors used symbols (a), (f), and (g) (p.1, line 33- 34). Are they (a), (b), and (c)? Please confirm them.
Author Response
Responses to Reviewer 3
Comments and Suggestions for Authors
This study examined the factors associated with fear and perceived risk of financial fraud against older people. The financial fraud against older people is one of the most important problems because the number of older people is increasing especially in Asia. Therefore, this study is valuable. However, I have some comments. The introduction is relatively long. Authors had better describe briefly in introduction.
This suggestion is well taken. A more concise introduction was offered (see pages 1 and 2)
On the other hand, the discussion is relatively short. There are few citations in the discussion and implications.
Thank you for the suggestion. We expanded our discussion and added perspectives based on some updated relevant studies. We put:
“There are four possible explanations for this outcome… Third, the study may have omitted the cognitive and psychological aspects of vulnerability. A recent study concluded that older adults with mild cognitive decline had higher scam vulnerability whereas those with moderate to severe cognitive decline had lower fraud vulnerability [37]. Another study demonstrated ....” (see page 10).
We also put additional relevant literature in the conclusion, including:
This speculation is supported by a previous study in which men were found to be more vulnerable to fraud but less likely to report actual victimization [37]. Lastly, no specific attention being given to digital fraud seems to be a limitation in this study. With the fast-growing number of internet and mobile phone users, the possible encounters between vulnerable targets (including elderly people) and scammers are also increasing. Internet users and consumer fraud has drawn the attention of researchers [43]. In future research, a more comprehensive framework with the incorporation of digital and no-digital fraud and cognitive factors such as age-related dementia, memory, and executive functioning [44] is worth considering. (see page 11).
I think that fear of financial fraud prevents the fraud victimization because people with high fear of financial fraud behave more carefully. In this study, female was more likely to be afraid of any fraud victimization than male. On the other hand, a recent study showed that male was more vulnerable to financial fraud (Ueno D, et al., Mild Cognitive Decline Is a Risk Factor for Scam Vulnerability in Older Adults. Front. Psychiatry, 20 December 2021. https://doi.org/10.3389/fpsyt.2021.685451.2021).
Thank you very much for the great suggestion. In the revised manuscript, we made good use of this reference to strengthen our perspective about fraud vulnerability (see page 11, and also our response to the previous comment).
Authors had better discuss the relationship between fear of financial fraud and vulnerability to financial fraud.
This point is well noted. We speculate that fear of financial fraud and vulnerability to financial fraud is not in a linear relationship. In reviewing previous literature, we noted cognitive function and depression may have an impact on this relationship. This point was not proved in our study. However, we proposed it in future study:
“A recent study concluded that older adults with mild cognitive decline had higher scam vulnerability whereas those with moderate to severe cognitive decline had lower fraud vulnerability [37]. Another study demonstrated that older victims of fraud compared with non-victims were more likely to exhibit severe depressive symptoms [38]. Therefore, ….” (see page 10).
The response rate of this study is low. Therefore, the participants in this study might not represent older people in Hong Kong. Please discuss this point.
Thanks for the comment. Our study adopted multistep sampling method so as to eliminate the distribution bias and to include the respondents with characteristics relevant to the topic of study. We added our elaboration below:
“A multistep sampling method was employed. It is expected that this sampling method can help minimize the distribution bias by controlling the community characteristics, service unit natures and respondent characteristics…Social service providers for physically active older adults and physically inactive ones in the selected districts were invited to distribute a letter of invitation to potential interviewees.” (see page 4).
I think that mental state, especially anxiety and depressive symptoms, affects the fear of fraud victimization and perceived risk of fraud victimization. Please discuss this point.
Thank you very much for the excellent suggestion. We added this point in the discussion part (see page 10, also our responses to a previous comment).
In introduction, authors used symbols (a), (f), and (g) (p.1, line 33- 34). Are they (a), (b), and (c)? Please confirm them.
Well noted. Correction was made accordingly. We put:
“Older people tend to become victims largely because they (a) are less likely to report and acknowledge being victims of financial fraud [4], (b) like keeping cash at home [5] , and (c) have a higher risk of developing dementia in later life [6]” (see page 1).
Round 2
Reviewer 3 Report
Most of issues suggested by reviewers were revised correctly. I think this manuscript is now suitable for publication.